# DEEP REINFORCEMENT LEARNING FROM HIERARCHICAL WEAK PREFERENCE FEEDBACK

## ABSTRACT

Reward design is a fundamental, yet challenging aspect of practical reinforcement learning (RL). For simple tasks, researchers typically handcraft the reward function, e.g., using a linear combination of several reward factors. However, such reward engineering is subject to approximation bias, incurs large tuning cost, and often cannot provide the granularity required for complex tasks. To avoid these difficulties, researchers have turned to reinforcement learning from human feedback (RLHF), which learns a reward function from human preferences between pairs of trajectory sequences. By leveraging preference-based reward modeling, RLHF learns complex rewards that are well aligned with human preferences, allowing RL to tackle increasingly difficult problems. Unfortunately, the applicability of RLHF is limited due to the high cost and difficulty of obtaining human preference data. In light of this cost, we investigate learning reward functions for complex tasks with less human effort; simply by ranking the importance of the reward factors. More specifically, we propose a new RL framework – HERON, which compares trajectories using a hierarchical decision tree induced by the given ranking. These comparisons are used to train a preference-based reward model, which is then used for policy learning. We find that our framework can not only train high performing agents on a variety of difficult tasks, but also provide additional benefits such as improved sample efficiency and robustness.

## 1 INTRODUCTION

Over the past decade, significant advancements in deep learning techniques, along with unprecedented growth in computational power, have facilitated remarkable achievements in the field of deep reinforcement learning (RL) across diverse domains, including finance, transportation, and automatic programming (Deng et al., 2016; Haydari & Yılmaz, 2020; Le et al., 2022). A key component of modern RL is the reward function, typically predefined in benchmark environments such as the OpenAI gym or games (Mnih et al., 2013; Silver et al., 2016; Brockman et al., 2016). When dealing with complex real-world environments, however, the absence of a predefined reward function necessitates explicit reward design based on human knowledge or implicit reward learning (Dewey, 2014; Sutton & Barto, 2018).

Handcrafting a reward function based on human knowledge poses significant challenges, as usually no single factor captures all aspects of agent behavior. Therefore, practitioners often use multiple reward factors $z_1, \ldots, z_m$, each of which describes different facets of the agent's behavior. These factors serve as the basic components for explicit reward design, and can usually be obtained with minimal computational cost. The most common approach to utilizing these factors involves linear or non-linear combinations (e.g., $r = \sum_i \omega_i * z_i$) (Booth et al., 2023; Le et al., 2022; Zhang et al., 2019). The hyperparameters $\omega_i$ are tuned to optimize the agent's task performance, a process commonly known as reward engineering (Fu et al., 2017; Wu et al., 2021).

Although reward factors may contain useful information, reward engineering is often not an optimal way to incorporate human knowledge about these factors. This is because reward engineering assumes the ground-truth reward can be formed as a simple combination of rewards factors, an assumption that may suffer from large approximation bias. Moreover, reward engineering usually requires a large tuning cost, as the optimal reward weights are not necessarily intuitive to human overseers and therefore must be chosen through trial and error. In particular, some reward fac-

tors may have different magnitudes as well as intricate dependencies with other reward factors. In this case, determining the weight from human prior knowledge becomes challenging, and multiple weights must be tuned at the same time since their respective reward factors are correlated.

To mitigate the aforementioned issues, researchers have explored alternatives for incorporating human knowledge into reward design. One straightforward approach is to employ human annotators to assign a reward to each agent trajectory (Knox & Stone, 2009; Pilarski et al., 2011; Griffith et al., 2013). However, this approach presents two major challenges: rewards assigned by different annotators exhibit significant variability, and rewards from the same annotators lack consistency. As a solution, reinforcement learning from human (preference) feedback (RLHF) has emerged, which employs annotators to rank agent trajectories. RLHF is currently one of the most promising approaches for reward design (Knox & Stone, 2008; Christiano et al., 2017; Stiennon et al., 2020; Ouyang et al., 2022). RLHF utilizes human preferences over a large set of state-action trajectories to train a reward model that mimics such preferences. Subsequently, the reward model is used to train RL agents that are aligned with human preferences. This preference-based reward modeling approach has demonstrated impressive performance and has yielded state-of-the-art chatbots like InstructGPT (Ouyang et al., 2022) and GPT-4 (OpenAI, 2023). Despite its efficacy and versatility, RLHF's practicality is constrained by the substantial cost associated with hiring human annotators.

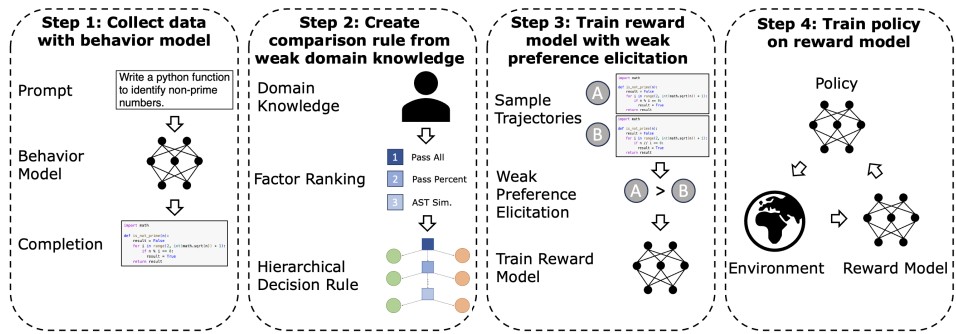

Figure 1: The HERON framework illustrated on the code generation task.

Seeking to improve upon the limitations of reward engineering and motivated by the recent success of RLHF, we propose to design reward functions based on a different form of domain knowledge: the importance ranking of each reward factor. Importance rankings of reward factors provide a natural means to express human domain knowledge, allowing practitioners to directly guide the agent in prioritizing different aspects of the task. To leverage these rankings effectively, we introduce a novel RL framework called HERON (**H**ierarchical preference-based w**E**ak **R**einf**O**rcement lear**N**ing). HERON compares RL trajectories using a hierarchical decision tree induced by the importance ranking. The decision tree first compares trajectories based on the most important factors and, if inconclusive, proceeds to assess less important factors. These comparisons are then used for training a preference-based reward model, which assigns higher rewards to trajectories that are favored more often. The reward model provides us a principled manner to convert rankings over the reward factors into a numerical reward. Ultimately, the reward function is used for policy learning. In contrast to reward engineering, HERON does not rely on the magnitude of each reward signal and enables the design of reward functions without specifying numerical weights for each reward factor. This approach not only significantly reduces the tuning cost but also improves the accessibility of the reward design process for individuals without RL expertise. Despite relying on this weaker form of domain knowledge (hence the weak RL designation), HERON's combination of hierarchical comparison and reward modeling allows for the creation of more complex rewards compared to those designed through reward engineering. An illustration of HERON can be found in Figure 1.

We empirically validate the HERON framework through extensive experiments on classic control tasks, robotic control, multi-agent traffic light control, and large language model fine-tuning for code generation (Brockman et al., 2016; Coumans & Bai, 2016; Le et al., 2022). In the control and traffic light environments, HERON consistently outperforms or achieves comparable performance to policies trained with reward engineering techniques, despite only utilizing the importance ranking of the reward factors as input. Regarding the code generation task, HERON demonstrates the ability to achieve higher Pass@$K$ scores compared to the hand-crafted rewards employed in state-of-the-

art approaches. Notably, we observe that HERON exhibits robustness to changes in the training environment, as it does not rely on the scale of individual reward factors, in contrast to reward engineering-based training.

Our contributions can be summarized as follows: (1) We propose the incorporation of a new form of domain knowledge – rankings over the importance of reward factors – into reward design; (2) We develop an algorithm called HERON, which utilizes a hierarchical comparison rule to effectively learn from such weak domain knowledge; (3) We validate the efficacy of HERON in various reinforcement learning environments, demonstrating that it not only matches the performance of reward engineering but also provides additional benefits such as improved sample efficiency and robustness.

## 2 RELATED WORK

Besides reward engineering and RLHF, there are several works that attempt to improve reward design for RL.

**Reward Shaping.** Reward shaping aims to accelerate the convergence of RL algorithms by incorporating auxiliary reward information through shaping functions (Ng et al., 1999; Tenorio-Gonzalez et al., 2010; Devlin & Kudenko, 2012). These approaches typically aim to mitigate the sparsity of a pre-defined reward function. While reward shaping has demonstrated success in practice, it often necessitates extensive reward engineering. To circumvent the need for costly tuning, several methods have been proposed to automatically shape rewards by utilizing an abstract MDP (Marthi, 2007), tile coding (Grzes & Kudenko, 2008), and bi-level optimization (Fu et al., 2019; Hu et al., 2020). In contrast, our work pursues a different direction that eliminates the requirement for a pre-specified reward function and does not assume that the reward is a linear combination of auxiliary factors.

**AutoRL.** AutoRL (Afshar et al., 2022; Parker-Holder et al., 2022) automates various aspects of hyperparameter selection in RL, including parameters related to the reward. Particularly relevant to our work, Faust et al. (2019) and Chiang et al. (2019) treat reward weights as hyperparameters and optimize them using population-based training.

**Weakly Supervised Reinforcement Learning.** In recent years, researchers have started investigating weakly supervised RL approaches, where only partial information about the reward is provided (Li & Yang, 2021; Cao et al., 2021; Kuutti et al., 2021). Our work shares similarities with the study by Lee et al. (2020), as both approaches learn from weak reward signals. However, there are large distinctions in both the goal and methodology: Lee et al. (2020) aims to leverage simple weak signal comparisons for faster learning of new tasks, while our work focuses on learning a reward from weak domain knowledge using more sophisticated hierarchical comparisons.

**Inverse Reinforcement Learning.** Inverse reinforcement learning (IRL) aims to learn a reward function from expert demonstrations (Ng et al., 2000; Abbeel & Ng, 2004; Boularias et al., 2011). Although IRL enables the learning of complex behaviors without manual reward tuning, it requires observed, optimal behavior. These demonstrations are often costly to obtain, and in our experiments, acquiring them would be far more expensive than obtaining a hierarchy of weak reward signals. Furthermore, IRL methods typically require unstable bi-level optimization procedures, which our approach does not involve.

## 3 METHOD

### 3.1 PROBLEM SETUP

We consider a Markov decision process $\langle \mathcal{S}, \mathcal{A}, \mathcal{P}, \mathcal{R}, \gamma \rangle$ where an agent interacts with an environment over a series of discrete time steps. At time step $t$, the agent receives an observation $s_t \in \mathcal{S}$, takes an action $a_t \in \mathcal{A}$, and receives the next state observation $s_{t+1} \in \mathcal{S}$ and reward $r_t \in \mathbb{R}$. The goal of the agent is to find a policy that maximizes the discounted sum of the reward, $\sum_{t \geq 0} \gamma^t r_t$.

Rather than assuming that we receive a reward at each time step, we instead consider a more realistic setting where a set of $n$ reward factors $z_t^1, \ldots, z_t^n$ are given at each time step. This setting is highly relevant to real-world applications where the ground truth reward function is unknown, but multiple measurements of a trajectory's quality can be obtained. We denote segments of the resulting trajectory as $\tau = (s_t, a_t, z_t^1, \ldots, z_t^n), \ldots, (s_{t+k}, a_{t+k}, z_{t+k}^1, \ldots, z_{t+k}^n)$. Note that we overload the notation for $z_i$ such that $z_i(\tau) = \sum_{(s_t, a_t) \in \tau} z_i(s_t, a_t)$. Similar to the RLHF approach, our goal is to

learn policies that act according to human preferences (Christiano et al., 2017). In our experiments, we may measure the performance of our algorithms by the ground truth reward (if one is known) or by alternative metrics not given during training.

## 3.2 ALGORITHM

The HERON algorithm trains policies based on human rankings over the reward factors. This is accomplished by iteratively carrying out the following four steps: (1) collecting trajectory data through policy rollouts, (2) eliciting weak preferences, (3) reward learning, and (4) policy learning.

**Data Collection.** The first step of the HERON algorithm is collecting trajectory data with a properly selected behavior policy. The collected data will later be used to train the reward model. There are several options for obtaining the behavior policy: For tasks such as code generation where expert demonstrations data are available, behavior cloning can be employed to acquire the behavior policy; When a handcrafted reward is available, it can be used to pre-train a behavior policy; The behavior policy can also be a purely randomized policy if no other options are available. However, using a purely randomized policy may introduce significant sampling bias due to the substantial discrepancy between it and the optimal policy. To address this issue, we employ multi-stage training as a means to enhance policy learning, which is further explained in Section 4.

**Weak Preference Elicitation.** To train the reward model over the collected data, HERON first compares trajectories based on an intuitive form of domain knowledge: rankings over the reward factors. In particular, we assume $z_1, \dots z_n$ have been ordered in descending order of importance by an expert with domain knowledge. We then elicit a preference $\mu \in \{0, 1, 2\}$ between trajectory pairs $(\tau_1, \tau_2)$ with a bifurcating decision tree induced by the given reward factor hierarchy. A tie is denoted by $\mu = 0$, $\mu = 1$ means $\tau_1$ is preferred, and $\mu = 2$ means $\tau_2$ is preferred.

The decision tree is constructed as follows. We first set the current level $l = 1$. We then calculate

$$\mu = \begin{cases} 0 & \text{if } |z_l(\tau_1) - z_l(\tau_2)| \leq \delta_l \\ 1 & \text{if } z_l(\tau_1) > z_l(\tau_2) + \delta_l \\ 2 & \text{if } z_l(\tau_2) > z_l(\tau_1) + \delta_l, \end{cases}$$

where $\delta_l$ is a margin hyperparameter for level $l$. The margin parameter $\delta_l$ is important since it ensures that we only elicit a preference using $z_l$ if the two trajectories are significantly different according to $z_l$. The margin $\delta_l$ can be used to inject further domain knowledge into the HERON algorithm, but in our experiments we set $\delta_l$ to the standard deviation of $z_l$ over the collected data.

If $\mu = 0$, we update $l \leftarrow l + 1$ and compare the trajectories with the next most important reward factor. If the two trajectories are not significantly different in any of the reward factors (i.e. $l > n$), we discard the trajectory pair. In this way a trajectory is preferred over another trajectory if it receives a significantly higher or similar scores on the reward factors that are most important to the human overseer. Our complete preference elicitation algorithm can be found in Algorithm 1 along with an accompanying diagram in Figure 2.

**Reward Learning.** Given a labeled dataset $D$ of trajectories $(\tau_w, \tau_l)$ where $\tau_w$ is the trajectory preferred by the weak preference elicitation algorithm (i.e. $\mu = 1$), we would like to assign a higher reward to the preferred trajectory (we remove all ties from the dataset, since we find including them has negligible effect on training). To accomplish this, we train a reward model $R_\phi : \mathcal{S} \times \mathcal{A} \rightarrow \mathbb{R}$ where $R_\phi(\tau) = \sum_{(s_t, a_t) \in \tau} R_\phi(s_t, a_t)$. To assign a higher reward to the preferred trajectory $\tau_w$, we follow the methodology in Ouyang et al. (2022) and optimize the loss

$$\mathcal{L}(\phi) = -\mathbb{E}_{(\tau_w, \tau_l) \sim D} \left[ \log \left( \frac{1}{1 + \exp(-(R_\phi(\tau_w) - R_\phi(\tau_l)))} \right) \right].$$

We remark that this loss essentially employs the Bradley-Terry preference model (Bradley & Terry, 1952). Once we have trained the reward model $R_\phi$, we can assign a reward to each trajectory $\tau$ as $R_\phi(\tau)$.

To enhance flexibility and further incorporate domain knowledge into the reward, we propose multiplying the reward $R_\phi(\tau)$ by a shaping constant, denoted as $\alpha^{F(\tau)}$. Here, $\alpha$ represents a hyperparameter, while $F(\tau)$ corresponds to the average level of the decision tree the trajectory $\tau$ wins at. By tuning $\alpha$, we can effectively control the reward's shape and the degree of separation between the

best and worst trajectories. A deeper discussion and visualization of this shaping constant can be found in Appendix I.1. We focus our $\alpha$ tuning efforts exclusively on the code generation task due to its higher complexity compared to the other tasks under consideration.

**Policy Learning.** Using the reward $R_\phi$ we can train a policy $\pi_\theta : \mathcal{S} \times \mathcal{A} \to \mathbb{R}$ to optimize the expected discounted return,

$$J(\theta) = \mathbb{E}_{\tau \sim \pi_\theta} \left[ \sum_{(s_t, a_t) \in \tau} \gamma^t R_\phi(s_t, a_t) \right].$$

This objective can be optimized via popular reinforcement learning algorithms such as Q-learning or Proximal Policy Optimization (Sutton & Barto, 2018; Schulman et al., 2015; 2017).

---

**Algorithm 1** Weak Preference Elicitation

**Input:** $n$ ranked reward factors $\{z_1, \ldots z_n\}$, margins $\{\delta_1, \ldots \delta_n\}$, and trajectory pair $(\tau_1, \tau_2)$.
**Output:** $\mu \in \{0, 1, 2\}$.
$l \leftarrow 1$ , $\mu \leftarrow 0$
**while** $l \leq n$ and $\mu = 0$ **do**
    **if** $z_l(\tau_1) > z_l(\tau_2) + \delta_l$ **then**
    $\mu \leftarrow 1$
    **else if** $z_l(\tau_2) > z_l(\tau_1) + \delta_l$ **then**
    $\mu \leftarrow 2$
    **else**
    $l \leftarrow l + 1$
    **end if**
**end while**

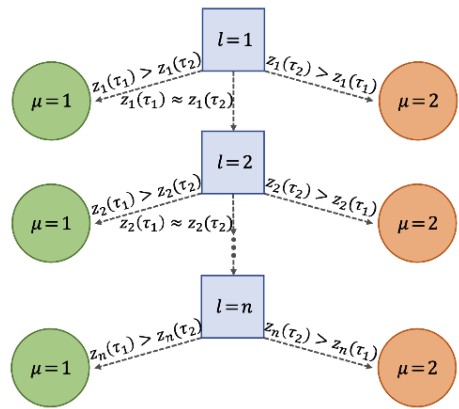

Figure 2: Weak preference elicitation.

---

## 4 EXPERIMENTS

We investigate the efficacy of our algorithm in a variety of single and multi-agent RL environments, including large scale language model finetuning.

### 4.1 ENVIRONMENTS AND BASELINES

**Classic Control.** We test HERON on two classic control environments: pendulum and continuous mountain car. In these environments a ground truth reward has been constructed as a linear combination of factors such as the location, force applied, and angle. To rank these reward factors for HERON, we simply use the magnitude of the weight of each factor. In this setting, we use DDPG as the RL algorithm and parameterize the reward model with a three layer MLP. For this task the behavior policy is randomly initialized, which as we mentioned before may lead to a large sampling bias for reward model training. We therefore adopt a multi-stage training approach, where the reward model is continually updated using trajectories collected from the most recent policy. Exact training details can be found in Appendix B. We evaluate over 10 random seeds.

**Robotic Control.** To test if HERON can be applied to more complex settings, we evaluate it on three robotic control tasks: Ant, Half-Cheetah, and Hopper. We use the PyBullet simulator, where the ground-truth reward is formulated as a linear combination of several factors such as the robot's potential, the power cost, whether the joints are at their limit, and whether the robot's feet collide (Coumans & Bai, 2016). More details on this environment can be found in Appendix K.

**Multi-Agent Traffic Light Control.** In this scenario, cooperative agents learn to increase the throughput and minimize the wait of cars passing through a traffic network. For the ground truth reward, Zhang et al. (2019) designs the reward as a linear combination of the following six weak reward factors: queue length, number of vehicles in each lane, average vehicle wait time, average vehicle delay, number of vehicle emergency stops, number of light phase changes, and number of vehicles passing through the system.

We use the same weak reward factors to build a variety of reward hierarchies. Our reward model is parameterized by a three-layer MLP that is learned by multi-stage training. We use QCOMBO (Zhang et al., 2019), a Q-learning based algorithm as the RL algorithm and conduct experiments

using the Flow framework (Wu et al., 2017). We train the MARL policies on a two-by-two grid (four agents), each parameterized by a three-layer MLP. For more details on the environment and experimental setting, see Appendix E.

In this setting we evaluate the learned policies based on the reward tuned in Zhang et al. (2019) and by examining the different weak reward signals. We remark that in traffic light control there is no one optimal reward function, and different reward functions may be preferred in different scenarios.

**Code Generation.** RL has recently gained considerable attention for its state-of-the-art performance in various text generation tasks. Therefore, we investigate if HERON can achieve similar improvements in LLM performance solely based on rankings over weak reward signals. In particular, we consider the code generation task.

In the code generation task, the goal of the agent is to write a program that will satisfy the criteria specified in a given problem. Recently, Le et al. (2022) demonstrated state-of-the-art performance can be achieved by training with RL. Le et al. (2022) (CodeRL) use a reward engineering approach where a weighted combination of reward signals including whether a program passes the unit tests and the type of program error is used as the reward. Shojaee et al. (2023) (PPOCoder) build upon this work, integrating more reward factors such as a program's abstract syntax tree (AST) similarity to expert demonstrations. Our decision tree is based on three factors: whether a program passes all tests, the percent of tests a program passes, and the AST similarity to expert demonstrations.

To train policies we follow the implementation of Le et al. (2022). We initialize our policies with the CodeT5plus-large model and our reward model with CodeT5-small (Wang et al., 2021). The policies are first trained with behavior cloning on the expert demonstrations. Next, we generate 20 samples per training program, and conduct RL training over these generated samples. We train with the policy gradient objective. We evaluate the performance of each algorithm using the pass@K metric, which is the number of programming problems passed with K submissions per problem (Chen et al., 2021). We primarily evaluate HERON on APPS, a python programming datasets containing 5000 test problems (Hendrycks et al., 2021). Each question in the dataset comes with expert demonstrations and test cases the program should pass. To evaluate each algorithm, we generate 200 programs per problem. In total, each method is evaluated on 1 million generated programs. To evaluate the generalization ability of the policies, we evaluate each policy in a zero-shot manner on the MBPP dataset, which contains 974 basic python programming questions (Austin et al., 2021).

**Baselines.** To demonstrate the efficacy of HERON, we compare the HERON framework with policies trained on the ground truth reward, a reward engineering baseline, and two ensemble-based baselines inspired by Brys et al. (2017). The ensemble approaches train a separate policy on each reward factor and then select an action at each timestep by a weighted combination of each policy. Formally, given a set of policies $\pi_1, \ldots, \pi_n$, we select an action at each timestep as $a \leftarrow \text{argmax}_{a \in \mathcal{A}} \sum_{k=1}^n w_k \pi_k(s, a)$, where $w_k$ is a weight to be chosen. We use two variants of this algorithm: a linear variant, where $w_k = \frac{1}{n}$ and a preference based variant, where $w_k = v^k$ ($\pi_k$ is trained on the $k$th most important factor) and $v \in [0, 1]$ is a hyperparameter to be tuned. For the reward engineering baseline, the reward is formulated as $\sum_{i=1}^n \beta^i z_i$, where $\beta$ is a hyperparameter selected from $\{0.3, 0.4, ..., 0.9, 1.0\}$ and $z_i$ are the normalized reward factors. The reward factors are ordered according to the HERON reward hierarchy, making this a very realistic and competative reward engineering baseline. See Appendix C for detailed description and evaluation of the baseline algorithms. For the code generation setting, we use the carefully tuned CodeRL and PPOCoder rewards as reward engineering baselines.

We evaluate the performance of each policy based on the reward engineering reward received for the control environment and by Pass@K for the code generation experiments.

## 4.2 MAIN RESULTS

**Control Environments.** The results for the pendulum and continuous mountain car control can be found in Figure 3. From Figures 3a and 3b, we can see that policies trained with HERON algorithm achieve comparable performance to policies trained with the ground truth reward. The results for robotic control can be found in Table 3, where we observe that although HERON cannot always perform as well as the ground truth reward, it can always exceed the performance of the reward engineering baseline. We remark that there is no clear reward hierarchy in the classic control or robotics environments, so in this case we mainly want to evaluate if HERON can be used to train

a reasonable policy. Together, these results support our claim that high performing policies can be trained using only rankings over reward factors.

**Multi-Agent Traffic Light Control.** This setting is a perfect environment to evaluate HERON in, as there are many reward factors. In Figure 3c, we plot the evaluation reward of policies in the traffic light control environment. We observe that the policy trained with HERON perform significantly better than the policies trained with the reward engineering baseline or by the ground-truth reward developed in Zhang et al. (2019). We hypothesize that HERON can utilize each reward signal better than a linear combination does; a significant change in a single reward factor may be drowned out in the linear combination, but HERON can flexibly incorporate this information due to its hierarchical nature. In addition, HERON's smooth reward function (opposed to sparse rewards from raw reward factors) may be more conducive for learning, and therefore lead to higher reward.

The ensemble-based methods do not perform well in any of the environments, as the policies trained on some of the less important reward factors will not learn information useful for completing the task. Even if we combine the policies according to the importance ranking, the ensemble-based baselines cannot compete with HERON.

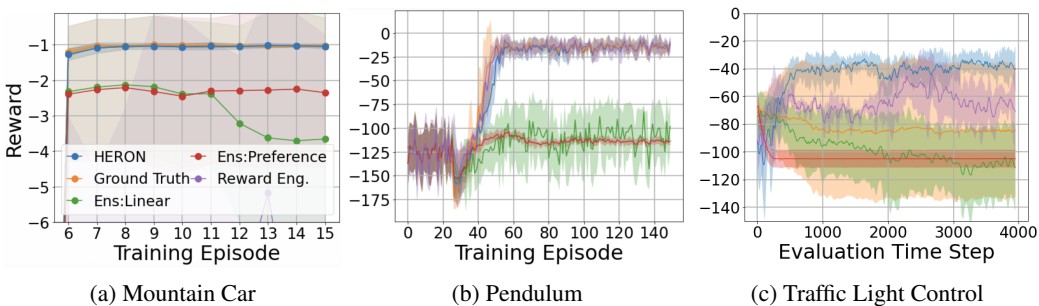

|          (a) Mountain Car          |          (b) Pendulum          |          (c) Traffic Light Control          |

Figure 3: Training and evaluation curves on different tasks (mean $\pm$ one standard deviation).

**Code Generation.** We display the results for the code generation task in Table 1. We also show performance across different question types in Table 4. HERON outperforms all other approaches. This is most likely because reward engineering only gives a large reward to programs that pass all the unit tests or are similar to the expert demonstrations, while HERON can give a large reward to programs that may fail some unit tests but the reward model predicts as being likely to satisfy the prompt. This means that HERON will promote a more diverse set of programming strategies. In addition, the smoothness of HERON's reward function may benefit policy optimization.

Table 1: Raw Pass@K on APPS.

|                          | Pass@1 | Pass@5 | Pass@10 | Pass@20 | Pass@50 |
|--------------------------|--------|--------|---------|---------|---------|
| Behavior Cloning (770M)  | 1.59   | 3.82   | 5.19    | 6.74    | 6.74    |
| CodeRL Reward            | 1.71   | 4.12   | 5.57    | 7.26    | 9.81    |
| PPOCoder Reward          | 1.23   | 3.08   | 4.19    | 5.50    | 7.62    |
| HERON                    | **1.72** | **4.19** | **5.71** | **7.49** | **10.19** |

Table 2: Filtered Pass@K on APPS.

|                   | Pass@1 | Pass@10 | Pass@20 |
|-------------------|--------|---------|---------|
| Behavior Cloning  | 4.70   | 6.36    | 6.44    |
| CodeRL Reward     | 5.73   | 8.57    | 8.96    |
| PPOCoder          | 5.60   | 8.61    | 8.93    |
| HERON             | **5.74** | **9.03** | **9.43** |

Table 3: Ground-truth reward obtained in robotics environments.

|              | Ant          | Hopper       | Cheetah      |
|--------------|--------------|--------------|--------------|
| Ground-truth | 0.99(0.0)    | 0.86(0.01)   | 0.94(0.10)   |
| Reward Eng.  | 0.88(0.02)   | 0.72(0.05)   | 0.61(0.04)   |
| HERON        | **1.0(0.01)** | **0.78(0.04)** | **0.62(0.04)** |

We further analyze code generation performance using the filtered Pass@$K$ metric, which only submits programs that pass unit tests provided in the prompt (Chen et al., 2021). As seen in Table 2, HERON uniformly outperforms the baselines, confirming the efficacy of HERON.

As in Le et al. (2022), we evaluate the performance of policies trained by HERON on the MBPP dataset. The results are displayed in Table 5. We display results for small $K$ only as larger $K$

Table 4: Pass@50 on APPS

|  | Intro. | Interview | Comp. |
| --- | --- | --- | --- |
| Behavior Cloning | 18.8 | 4.23 | 2.10 |
| CodeRL Reward | 23.7 | 6.93 | 4.51 |
| PPOCoder Reward | 18.6 | 5.41 | 3.23 |
| HERON | **24.6** | **7.28** | **4.53** |

Table 5: Pass@K on MBPP

|  | Pass@1 | Pass@2 | Pass@5 |
| --- | --- | --- | --- |
| CodeRL Reward | 6.58 | 10.27 | 16.24 |
| PPOCoder Reward | 6.58 | 10.09 | 15.85 |
| HERON | **7.40** | **11.03** | **16.54** |

may result in high variance on the smaller MBPP dataset. HERON outperforms the other methods, indicating that HERON can result in generalizable policies.

### 4.3 FLEXIBILITY OF HIERARCHICAL REWARD MODELING

In various tasks, there is no one ideal reward, and the aspects of an agent's behavior that should be prioritized depend on the practitioner's preference. As a result, a crucial characteristic that reward design algorithms should possess is flexibility. In particular, modifying the domain knowledge inputted should result in corresponding changes in the behavior of the agent. To evaluate the flexibility of HERON, we examine how changing the reward factor rankings changes agent behavior in the traffic light control environment.

In this experiment we always set the most important factor as the number of cars passed, and then we use the queue length, wait time, or delay as the second factor. The performance of HERON policies trained with these reward hierarchies can be seen in Figure 4. All of the policies have a similar number of cars passing through the system, which makes sense since this factor is prioritized. However, for wait time, only the policy that has wait time ranked second maintains stable performance. Similarly, for delay and queue length the policies that prioritize these factors achieve better performance than the policy that prioritizes wait time. From these experiments we can observe that HERON is quite flexible, and that by changing the reward hierarchy we can significantly influence the agent's behavior. The results with more hierarchies can be found in Appendix G. To further increase the flexibility of HERON, we could tune the margin parameter of each reward factor, therefore impacting the fraction of decisions made at each level of the decision tree.

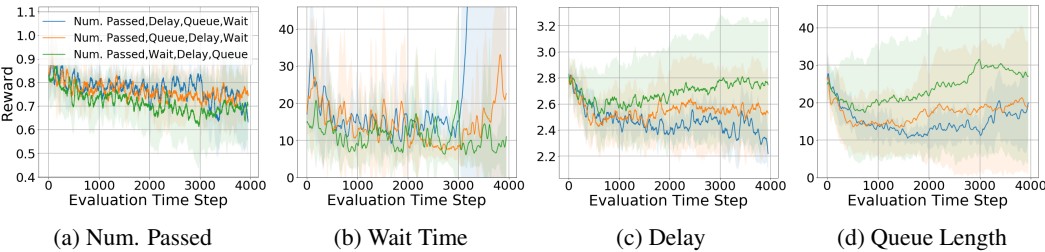

| (a) Num. Passed | (b) Wait Time | (c) Delay | (d) Queue Length |
| --- | --- | --- | --- |

Figure 4: Evaluation curves with different reward hierarchies (mean $\pm$ one standard deviation).

### 4.4 ROBUSTNESS

A subtle advantage of HERON is that unlike reward engineering, it does not depend on the magnitude of the different reward factors. This is because the weak preference elicitation algorithm will label trajectory pairs with $\mu \in \{0, 1, 2\}$, regardless of the scale of the different factors. This scale-invariance is beneficial, since algorithms that do depend on the scale of the reward factors may be vulnerable to changes in the environment during training. For example, if the scale of a reward factor suddenly doubles, (i.e. the traffic on a highway doubles due to rush hour) then two things will happen: (1) the scale of the reward signal may sharply increase, which is similar to a sudden change in learning rate (2) the weight vector used in reward engineering to combine the reward factors will effectively be changed. The first phenomenon may cause training instability, and the second phenomenon could cause the agent to be misaligned with the human overseer's desires.

We hypothesize that HERON's invariance to the reward scale will provide robustness against changes in the training environment. To test this hypothesize, we change the speed of the cars halfway into training (this a realistic setting, since many areas have time-dependent speed lim-

its). We then evaluate each policy after training under the new environment, and see which algorithms were able to adapt the best. We compare the HERON-trained policy with two policies trained with the reward engineering: one that uses the optimal learning rate in the unchanged environment ($1 \times 10^{-3}$) and one that uses a smaller, more stable learning rate of $1 \times 10^{-5}$.

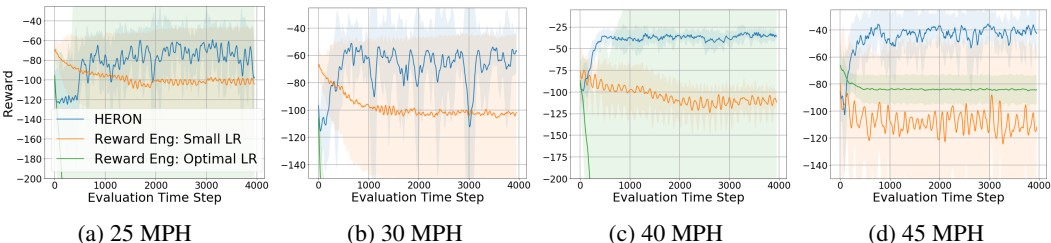

| (a) 25 MPH | (b) 30 MPH | (c) 40 MPH | (d) 45 MPH |

Figure 5: Evaluation curves with different speed changes during training (mean $\pm$ one standard deviation). The normal speed limit is 35 mph.

From Figure 5, we can see that reward engineering is quite sensitive to changes in the environment during training. This can be combatted with a smaller learning rate, but this will result in slower learning and a sub-optimal reward. On the other hand, HERON is able to attain a high reward regardless of the environment change, supporting our hypothesis that HERON's scale-invariant design leads to increased robustness.

## 4.5 ANALYSIS

**Training Time.** The main computational cost of HERON comes from the reward model training, as data collection is already a part of most RL algorithms and weak preference elicitation is very fast. When we do multi-stage training we can accelerate training by using an annealed training schedule (see Appendix I). The normalized training time of HERON, reward engineering, and ensemble-based learning can be found in Figure 6a. HERON is 25% slower than reward engineering on average, which is quite reasonable given that the tuning cost of reward engineering is usually large.

**Ablation.** Depending on the application, HERON introduces up to two hyperparameters, $\delta_i$ and $\alpha$. We only tune $\alpha$ on the code generation task, where the underlying reward may be very complex. We set $\delta_i$ to the standard deviation of $z_i$ over the collected data in our experiments. Nonetheless, we evaluate the sensitivity of HERON to these parameters in Figure 6. Tuning $\alpha$ can improve performance, but HERON is on par with reward engineering for all choices of $\alpha$. Appendix I.1 presents a plot illustrating the relationship between the reward shape and $\alpha$. For $\delta_i$, we find values in $[0, 2 * \sigma_i]$ work well, where $\sigma_i$ is the standard deviation of $z_i$.

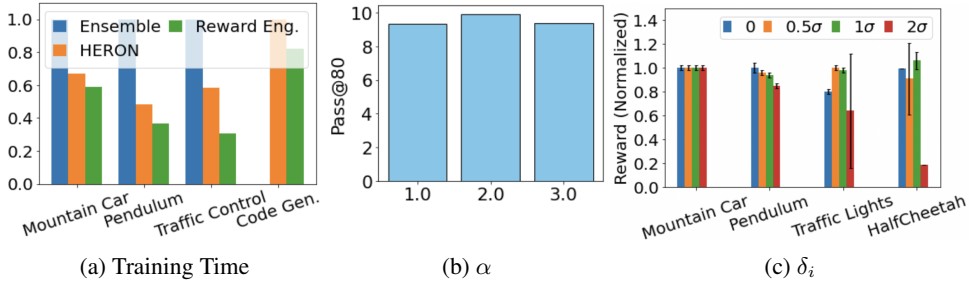

| (a) Training Time | (b) $\alpha$ | (c) $\delta_i$ |

Figure 6: Training time and ablation study for HERON.

## 5 DISCUSSION

In this work we propose HERON as an alternative to reward engineering. Which one can better capture the reward may be problem specific. In our experiments, we find that HERON usually performs better than reward engineering, while reducing the tuning cost. Another direction for future work regards multi-objective RL (MORL). MORL attempts to find an optimal policy for each linear combination of reward factors and is useful for settings where the agent needs to adapt to different rewards (Van Moffaert & Nowé, 2014; Yang et al., 2019). To adapt to changes in the weak domain knowledge, it would be would be interesting to extend HERON to learn a policy for each possible reward factor ranking in a similar manner to MORL.

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

## A    APPENDIX

## B    CLASSIC CONTROL EXPERIMENT DETAILS

For the classic control experiments we use the OpenAI gym (Brockman et al., 2016). To train all policies we use the DDPG algorithm, where the policies are parameterized by three layer MLPs with 256 hidden units per layer. We use the Adam optimizer, and search for a learning rate in $[1 \times 10^{-5}, 1 \times 10^{-3}]$.

For mountain car we train for a total of 15000 timesteps and begin training after 5000 timesteps. For pendulum, we train for a total of 50000 timesteps and begin learning after 25000 timesteps.

## C    BASELINES

Beyond the ground-truth reward, we compare the HERON algorithm with two ensemble baselines inspired by Brys et al. (2017). These ensemble baselines train a separate policy on each reward factor, and then combine the policies' outputs in a given state to select an action. In every environment we train each policy in the ensemble with the similar parameters as used for the reward engineering baseline and we again tune the learning rate in $[1 \times 10^{-5}, 1 \times 10^{-3}]$.

As described in the main text, we consider two variants of this ensemble based algorithm: one where the action is selected according to an average over each policy ($a \leftarrow \text{argmax}_{a \in \mathcal{A}} \sum_{k=1}^{n} \frac{1}{n} \pi_k(s, a)$) and one where the preference ranking used as input to HERON is used to combine the actions ($a \leftarrow \text{argmax}_{a \in \mathcal{A}} \sum_{k=1}^{n} \gamma^k \pi_k(s, a)$). With the second variant, $\gamma$ is selected from $\{0.25, 0.35, 0.45, \cdots, 0.95, 0.99, 1\}$.

We also examine the performance of a reward engineering baseline where the reward is formulated as $\sum_{i=1}^{n} \beta^i z_i$, where $\beta$ is a hyperparameter selected from $\{0.3, 0.4, ..., 0.9, 1.0\}$ and $z_i$ are the normalized reward factors. The reward factors are ordered according to the HERON reward hierarchy, making this a very realistic and competitive reward engineering baseline. However, we came across a few challenges when trying to make this algorithm work. First, the reward factors all need to be normalized, which either requires complex algorithms or multiple agent rollouts before training. In addition, we find that this baseline is very sensitive to $\beta$ and therefore has a higher tuning cost. In addition, it can often not beat the performance of HERON. We plot the performance of the reward engineering baseline in Figure 7. Note that this plot shows performance over all of training, and HERON typically displays larger reward (comparatively) in the last stages of training.

As we can see from Figure 7, the reward engineering baseline requires extensive tuning to achieve good performance. In addition, the choice of normalization strategy is very important (Figure 7f). These results further show the benefits of HERON.

## D    ROBOTICS

All of our experiments are conducted with the PyBullet simulator (Coumans & Bai, 2016). The reward factors in each environment are as follows: for Ant, it is whether the robot is alive, the progress towards the goal state, whether the joints are at their limits, and whether the feet are colliding. For HalfCheetah, the factors are the potential and the power cost. For Hopper, the factors are the potential, an alive bonus, and the power cost.

## E    TRAFFIC LIGHT CONTROL

In our experiments we train four agents in a two by two grid. The length of each road segment is 400 meters and cars enter through each in-flowing lane at a rate of 700 car/hour. The traffic grid can be seen in Figure 8. The control frequency is 1 Hz, i.e. we need to input an action every second. The reward is based on the following attributes for each agent $n$:

- $q^n$: The sum of queue length in all incoming lanes.
- $wt^n$: Sum of vehicle waiting time in all incoming lanes.

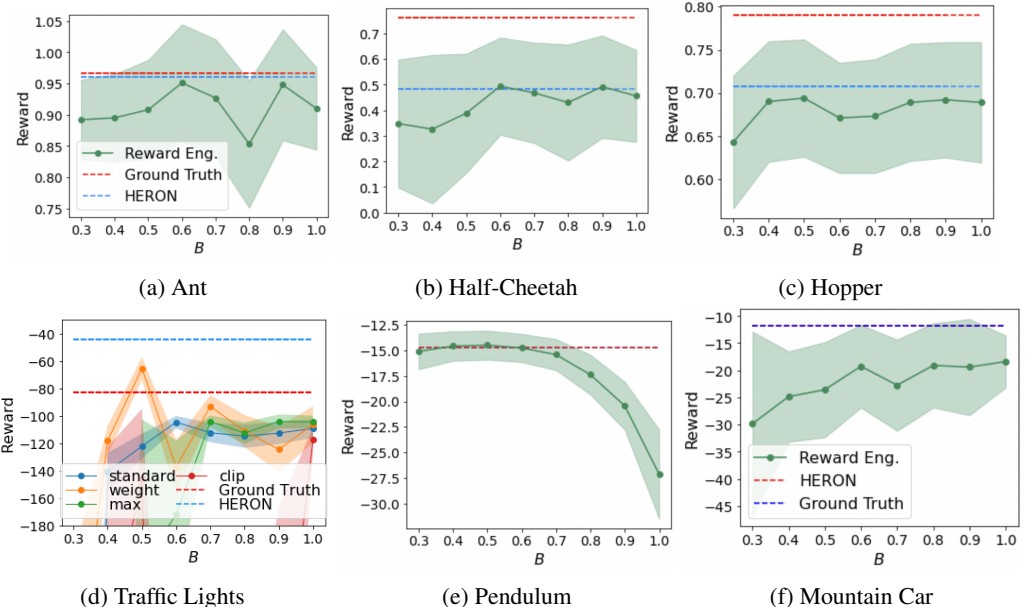

Figure 7: Ablation study of the reward engineering baseline.

- $dl^n$: The sum of the delay of all vehicles in the incoming lanes.
- $em^n$: The number of emergency stops by vehicles in all incoming lanes.
- $fl^n$: A Boolean variable indicating whether or not the light phase changed.
- $vl^n$: The number of vehicles that passed through the intersection.

We can then define the reward-engineering reward as

$$R^n = -0.5q^n - 0.5wt^n - 0.5dl^n - 0.25em^n - fl^n + vl^n.$$

All algorithms have the same training strategy. Each agent is trained for three episodes with 3000 SUMO time steps each. At the beginning of training the agent makes random decisions to populate the road network before training begins. Each algorithm is evaluated for 5000 time steps, where the first 1000 seconds are used to randomly populate the road. For adversarial regularization, we use the $\ell_2$ norm to bound the attacks $\delta$.

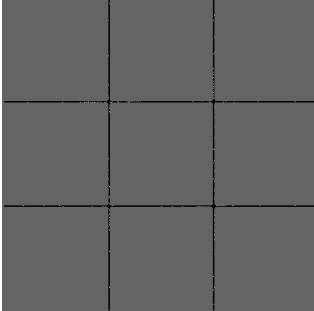

Figure 8: Traffic light control environment.

## F    RLHF COMPARISON

To explicitly compare RLHF with HERON, we compare the algorithms in the pendulum environment. To simulate human feedback, we rank one trajectory over another if the ground truth reward achieved by that trajectory is higher than the ground truth reward achieved by the other trajectory.

We then evaluate the performance of this simulated RLHF algorithm when varying amounts of feedback are given. The results can be seen in Figure 9. In this table we vary the number of feedbacks in RLHF, while keeping the number of feedbacks for HERON constant. In this setting HERON can perform as well as RLHF, but such good performance is not guaranteed in every environment.

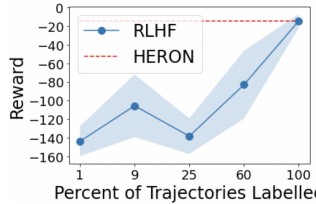

Figure 9: RLHF comparison in the Pendulum Environment.

## G    HERON FLEXIBILITY

In this section we evaluate how the behavior of the policies trained by HERON change when we change the reward hierarchy. We plot several hierarchies in Figure 10. The reward engineering is the thick black line. We try three factors as the most important factor (num_passed, wait time, and delay). We notice that all these observations can outperform the reward engineering reward, even though we measure the return with the reward engineering reward. One important deviation from this good performance is when wait time is not ranked highly. The wait time is a very important factor, and when we do not put this variable high up in the hierarchy, the performance becomes unstable when measured according to the reward engineering reward. This is because if we ignore the wait time of cars, the policy may make some cars wait for a long time, which is not ideal. However, this can easily be accounted for in the reward design process.

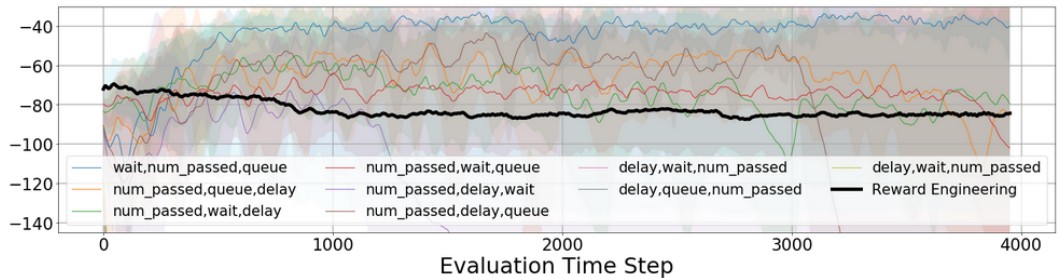

Figure 10: Different reward hierarchies in HERON.

We also show the level the decision tree induced by HERON reaches in Figure 11. This may change with different reward hierarchies (this one in particular priorities queue length, wait time, and delay), but as we can see from the figure, a relatively similar proportion of decisions are made at each level of the decision tree. We also remark different reward factors may be correlated (i.e. queue length and number passed), so the second factor may not have many decisions made with it.

## H    CODE GENERATION

In this section we describe details for the code generation task.

### H.1    BEHAVIOR CLONING

To train the initial behavior model we use behavior cloning (supervised fine-tuning) to adapt the pre-trained CodeT5 to the APPS task. In particular, we use train with the cross-entropy loss for 12000 iterations, using a batch size of 64. We use the Adam optimizer with a learning rate of $2 \times 10^{-5}$.

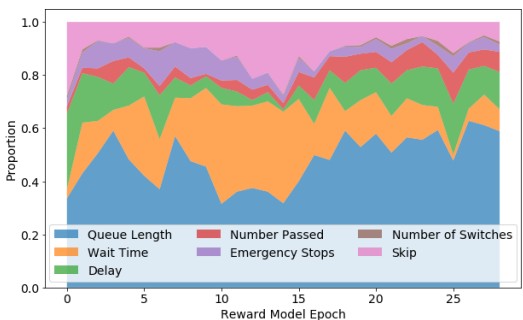

Figure 11: Different level reached by decision tree in HERON.

## H.2 TEMPERATURE SELECTION

A hyperparameter that can have a large impact on generation quality is the temperature parameter, which essentially alters how greedy we are in the next-token sampling step. In all settings we follow the implementation of Le et al. (2022), using a temperature of 0.6 for APPS and 1.2 for MBPP. In addition, we sample tokens greedily to construct a baseline sample for each problem.

## H.3 REWARD MODEL

It has been noted that reward models often overfit to the dataset (Ouyang et al., 2022). Therefore we use a smaller version of CodeT5 for our reward model with only 220 million parameters. We train this model for around 40000 steps with a batch size of 64. This is roughly a single epoch on the preference dataset, which is comprised of 20 samples per problem sampled from the behavior model and some expert samples provided by the APPS dataset. We use the Adam optimizer with a learning rate of $2 \times 10^{-5}$.

## H.4 REINFORCEMENT LEARNING

Once we have trained the reward model, we assign a reward to each program in our preference dataset and train using reinforcement learning on this dataset. Similar to Le et al. (2022), we train on the policy gradient loss and add the cross entropy loss as a regularization term. We compare our method to two reward engineering rewards:

**CodeRL reward.** The first reward we compare HERON to is from CodeRL, which defines the reward as

$$r_{\text{CodeRL}}(s) = \begin{cases} -1.0 & \text{if program } s \text{ fails to compile} \\ -0.6 & \text{if program } s \text{ has a runtime error} \\ -0.3 & \text{if program } s \text{ fails a unit test} \\ 1.0 & \text{if program } s \text{ passes all unit tests.} \end{cases}$$

**PPOCoder reward.** The second reward we compare HERON to is based on PPOCoder, which has the insight to include syntactic similarity to expert samples in the reward. This effectively smooths the reward, and can therefore make the reward more informative. In particular, they compare the abstract syntax trees of the generated programs with the expert example programs. This is computed as

$$R_{ast}(s, \widehat{s}) = Count(AST_s, AST_{\widehat{s}})/Count(AST_s).$$

We then construct the final PPOCoder based reward as $r_{\text{PPOCoder}}(s) = r_{\text{CodeRL}}(s) + \lambda MEAN_{\widehat{s}}(r_{ast}(s, \widehat{s}))$, where $MEAN$ is the mean operator. We tune $\lambda \in \{0.001, 0.01, 0.1, 1\}$. We remark that the original PPOCoder reward contains more reward factors, but we do not use all of them due to the large tuning cost required to tune the ourselves.

For both of these rewards and the HERON reward we tune the learning rate in $\{3 \times 10^{-6}, 5 \times 10^{-6}, 8 \times 10^{-6}\}$.

### H.5 Example Programs

To further analyze the performance of HERON, we examine some of the programs generated by HERON. These programs are randomly selected. We display concatenated prompts and completions in Figure 12.

## I  Reward Training

In this section we detail our reward model training. For the classic control tasks and the traffic light control task we do not have a good initial behavior policy, so we must train our reward model in an iterative manner. In these settings, we iteratively update the reward model using samples from the current version of the policy. In this way the reward model is trained on samples generated from progressively better policies.

As we mentioned in our discussion on the computational costs of HERON, the cost of reward model training depends on the frequency at which the reward model is trained. For the classic control environments we simply use a linear training schedule, in which the reward model is updated every 400 steps. For traffic light control we train the reward model with an annealed frequency, where the reward model is trained every $100v^t$ steps, where $v$ is set 1.3 and $t$ is the current time step.

We demonstrate the multi-step reward model training in Figure 13. The sharp drop in accuracy occurs at time step 1000, where the behavior model changes from random to a trained policy. This large change in accuracy indicates that multi-step reward model training is needed, as reward models trained on random behavior do not perform as well when the behavior changes.

### I.1  The $\alpha$ Hyperparameter

**Formal description of shaping factor:** Given a trajectory $\tau$, let us compare it with $n$ other trajectories $\tau_1, \ldots, \tau_n$. Let $F(\tau)$ denote the average level of the decision tree $\tau$ wins at. To allow us to incorporate domain knowledge into HERON, we multiply the reward assigned to $\tau$ by a factor $\alpha^{F(\tau)}$, where $\alpha$ is a hyperparameter. When the reward factors are categorical, $F(\tau)$ can capture which category $\tau$ lies in, and multiplying the reward by $\alpha^F(\tau)$ can control the reward separation between different categories.

**Visual description of shaping factor:**As mentioned in the main text, the $\alpha$ hyperparameter can be used to control the shape of the rewards. In Figure 14, we show how changing $\alpha$ changes the reward shape in the code generation task.

## J  Computational Setup

For the classic control tasks and traffic light control experiment we run experiments on Intel Xeon 6154 CPUs. For the code generation task, we train with Tesla V100 32GB GPUs.

## K  Robotics Learning Curves

In Figure 15 we display the learning curves in the robotics environments.

## L  Limitations

The main limitation of HERON is that not every problem will contain an obvious ranking over the reward factors, as some factors may be equally important. We propose to mitigate this limitation in future works by allowing for ties or using a randomized decision tree in the preference elicitation procedure.

```
You are an expert Python programmer, and here is your task: Write a function to find the list with minimum length us
ing lambda function. Your code should pass these tests:

assert min_length_list([[0], [1, 3], [5, 7], [9, 11], [13, 15, 17]])==(1, [0])
assert min_length_list([[1,2,3,4,5],[1,2,3,4],[1,2,3],[1,2],[1]])==(1,[1])
assert min_length_list([[3,4,5],[6,7,8,9],[10,11,12],[1,2]])==(2,[1,2])

class Solution:
        def min_length_list(self, *invars):
                """
                :type invars: List[List[int]]
                :rtype: (int, List[int])
                """
                res, index = float('inf'), []
                for item in invars:
                        if item == float('inf'):
                                return 1, index
                        elif not res:
                                index = index + [len(item)]
                        else:
                                if index[-1] < len(item):
                                        res, index = len(item), index + [len(item)]
                                else:
                                        index = index + [len(item)]
                return res, index
You are an expert Python programmer, and here is your task: Write a function to find the nth tetrahedral number. You
r code should pass these tests:

assert tetrahedral_number(5) == 35.0
assert tetrahedral_number(6) == 56.0
assert tetrahedral_number(7) == 84.0

class Solution:
        def tetrahedral_number(self, *n):
                print(n)

                def f(x, x1, x2, y):
                        if y < 0: return -1.0
                        x, y = x + x1, y + y1
                        if ((x > 5e-6 or y > 5e-6) and (x - x1 <= y - x2)): return -1.0
                        if (y == 5e-6 and x1 < y - x2) or y == 5e-7 and x1 == x2: return -1.0
                        z = 0
                        while (x > x1 <= x2):
                                if (x % 2 == 0): y -= 1
                                x = x // 2
                        while (y > y1 <= y2):
                                if (y % 2 == 0): x -= 1
                                y = y // 2
                        while (z > abs(y1 - x1) > abs(y2 - x2)):
                                z = abs(y1 - x1)
                                x1, x2, y1, y2 = x2, x1, y2, z * (-1)
                        return z * y

                a, b, w = *n
                p, q = 1, 3
                while True:
                        x, y = a * q + x, b * w + y
                        z = f(x, y, a * q, b * w + (a + 1) * q) if n else -1.0
                        if z < 0.0:
                                return y
                        if (a % 2 == 0): y = y - 1
                        a = a // 2
                        if (q < 2): a = p * q + q
                        if (p > 1.1e-6 and q > 5.6e-7): p = 1.1e-5
                        if (w > 1.2e-6): w = 1.2e-6
                        q = p * q + p

You are an expert Python programmer, and here is your task: Write a function to convert the given decimal number to
its binary equivalent. Your code should pass these tests:

assert decimal_to_binary(8) == '1000'
assert decimal_to_binary(18) == '10010'
assert decimal_to_binary(7) == '111'

class Solution:
        def decimal_to_binary(self, x):
                """
                :type x: int
                :rtype: str
                """
                a = [0]*8;
                for i in range(8):
                        a[i]+=1<<i;
                return ''.join(map(str, a));
```

Figure 12: Example programs generate by LLMs trained with HERON.

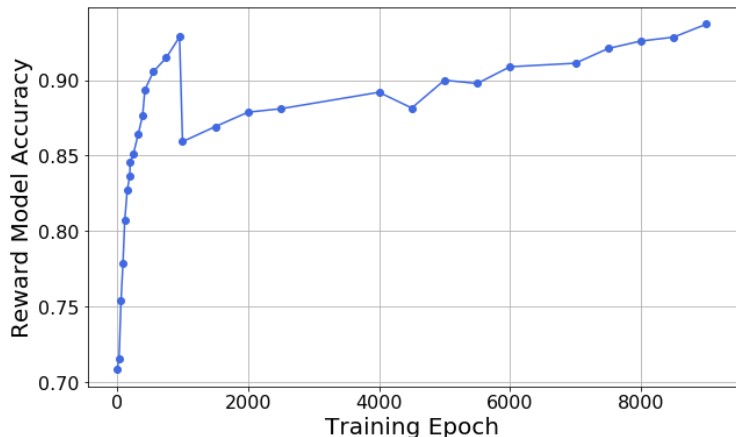

Figure 13: Reward model accuracy throughout training.

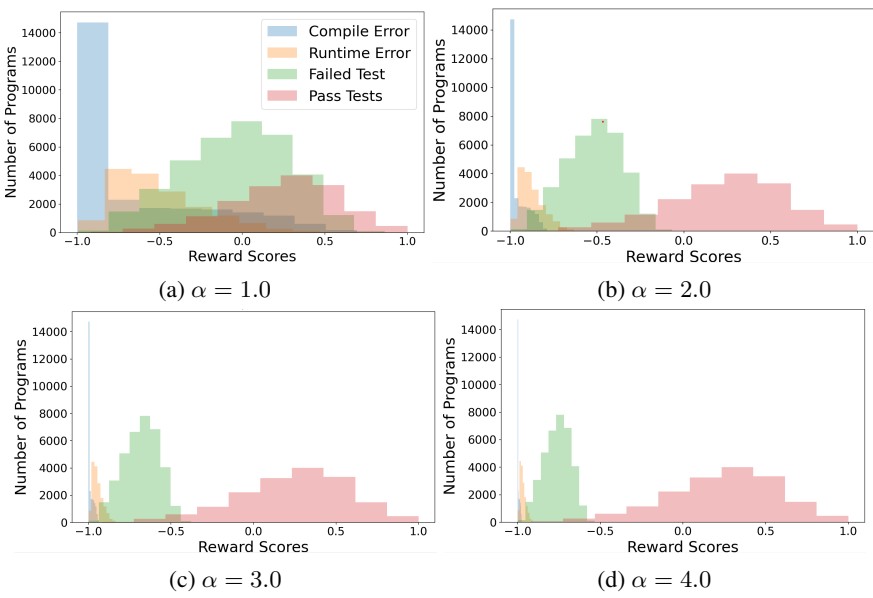

Figure 14: Reward shape with different values of $\alpha$.

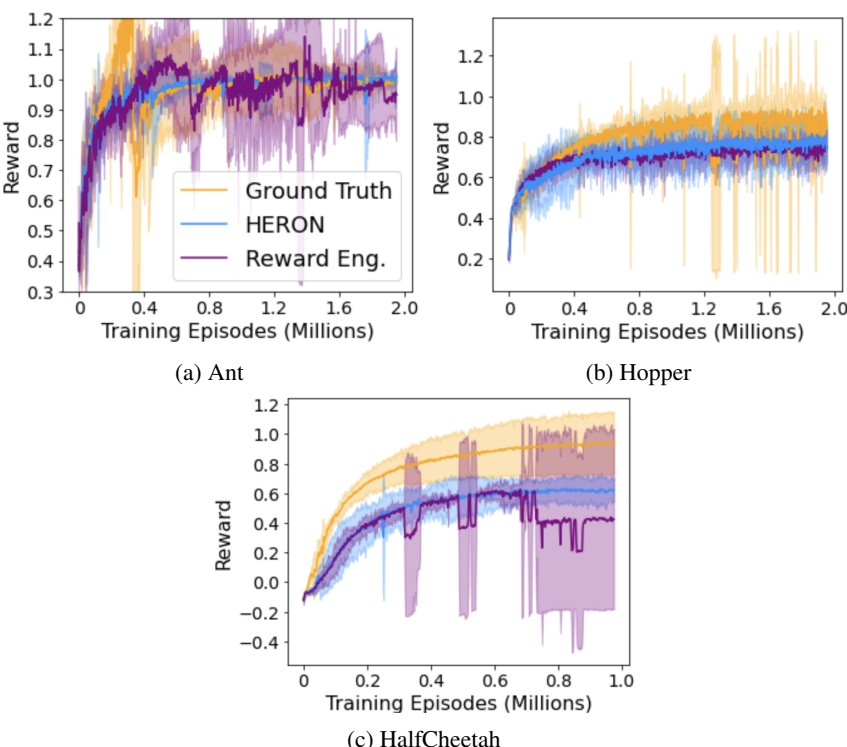

(a) Ant

(b) Hopper

(c) HalfCheetah

Figure 15: Training curves in different robotics tasks.

