# OpenReview forum: "Deep Reinforcement Learning from Weak Hierarchical Preference Feedback"
_ICLR.cc/2024/Conference — Submitted to ICLR 2024_

### Official Review · Reviewer_roR3 · 2023-10-27

**Soundness:** 2 fair
**Presentation:** 2 fair
**Contribution:** 2 fair
**Rating:** 3
**Confidence:** 3

**Summary:**

The authors propose a new method for reducing the cost of human annotations when trying to avoid manual reward engineering for reinforcement learning. Their method uses several easy to compute reward factors and it compares trajectories based on these factors by going from more important to a less important factor. These comparisons are then used to fit a reward function which is used in RL optimisation. The authors present results on a set of diverse benchmarks ranging from control to traffic simulation and coding.

**Strengths:**

- The paper attempts to design an alternative approach to reward engineering, which could potentially lead to more principle and stable performance.
- The method also avoids costly human annotations.
- The authors conduct a number of ablations and analysis targeted at understanding the strengths and weaknesses of the proposed method.
- The authors study a diverse set of applications from different domains.

**Weaknesses:**

- The paper is not very clearly written and it left me with a few unclear points which are important for the proposed problem setting. The authors start by talking about human feedback in reinforcement learning, but it is not clear from the method description what feedback is provided by humans. Is it the order of factors? Or is it feedback in the form of comparisons? What exactly is the problem setting? For example, what are the assumptions about the relationship between the reward factors and the true reward?
- Assuming that humans provide the comparisons in the form of weak preferences, it is not clear to me how it can be ensured that comparisons are in the form described on page 4 (fixed precise threshold, always correct). Also, usually in RLHF the humans would provide comparisons of full trajectories because this is a more reliable feedback compared to other types of feedback. Assuming that the comparisons are coming from the algorithm itself, it seems to me that the method is just a form of reward engineering that allows one to find the coefficients by ensuring certain rules (the importance order of factors) are respected.
- One limitation of the method is that it requires reward factors which are hand crafted. In many real world applications of RL there has been a tendency of moving away from the hand crafted rewards and learning reward functions directly from the observations which are represented in the raw form (e.g., images or text). I would like to hear a discussion on the applicability of the method to the various realistic domains.
- I have some questions and concerns about the experiments. For example, in all three environments the authors say that the ground truth reward is a linear combination of the factors. In this case it is not clear to me why one would need a non-linear function (MLP) for the learnt reward. Then, regarding the baselines, a classical baseline would be a (potentially linear) reward learnt as a function of factors on the basis of trajectory comparisons from human annotators. This is the most classical RLHF setting and it would be informative to compare the proposed method to it.

Minor points:
- I didn't find Algorithm 1 and Figure 2 necessary for understanding that part of the methodology, I think it can be skipped in favor of a more clear description of the problem setting and assumptions.
- I didn't understand why GT line is worse than other methods in Figure 3(c).
- Section 4.3 argues that the proposed method is quite flexible. It would be nice to hear a discussion of how this compares to the studied baselines such a reward engineering.

**Questions:**

I would like the authors to elaborate on the problem setting and assumptions (in particular relationship between the factors and true rewards). Also, I would like to understand better the principal difference of the proposed method to reward engineering and to see comparison to a more traditional reward learning scenario from RLHF.

---

> ### Author Response · Authors · 2023-11-20
> **Thank you for the helpful review!**
>
> Dear Reviewer roR3:
>
> Thank you for the insightful and critical review of our work. We believe that your first two cited weaknesses stem from a misunderstanding, and that by enhancing the clarity of our paper, they can be addressed. We also include discussions on your second two points, which we believe only require minor adjustment to be addressed in our paper. Please let us know if you have any further questions.
>
> **Weakness 1: Lack of clarity**
> - The general setting we consider here is one where at each time step, we receive a state observation, take an action, and then receive the next state as well as a list of reward factors. This setting is the same as that commonly used for reward engineering. Our setting is described in detail in Section 3.1 of the submitted manuscript.
> - Given this setting, HERON asks the human overseer to assign rankings over the order of the set of reward functions, $z_1, …, z_t.$ A concrete example would be (wait time > queue length > number of emergency stops) in traffic light control. This is described in the weak preference elicitation paragraph. These rankings are then used for learning.
> - We make the assumption that a decision tree over the reward factors will be able to approximate the true reward. Using this assumption has two main benefits: (1) constructing the decision tree requires less human input than the typical linear combination that reward engineering employs and (2) decision trees are highly flexible, and can therefore provide a good approximation of the true reward in most practical scenarios. Our experimental results back such an assumption.
>
> **Weakness 2: reliable comparisons**
> - In HERON, the comparisons are coming from the algorithm itself, and there is no human involved in the preference elicitation stage (see Algorithm 1).
> - Indeed, HERON is a new reward design algorithm, but we believe that it is a novel and significant contribution: no algorithms have been proposed to use the type of rankings that HERON uses, and HERON is a principled method that can outperform classical reward engineering approaches, only utilizing the rank of the reward factors and gaining robustness.
>
> **Weakness 3: Reliance on reward factors**
> - Thank you for bringing this point up. Indeed, HERON is designed for the classical setting where a set of reward factors is available. We believe that such a setting can still be useful, as demonstrated by our experiments in code generation, robotics, and traffic light control, all of which are practical environments where learning directly from human feedback would be extremely costly.
> - Moreover, we believe that using reward factors could boost emerging applications which require reward learning directly from the input, such as LLM finetuning. For example, in LLM finetuning, we could use tools such as grammar checkers and unit tests as the reward factors to allow HERON to be used. Such tools could even be used in conjunction with human feedback.
>
> **Weakness 4: Experiments and comparison to RLHF**
> - A key advantage of including the trajectory as the input of the reward model is that doing so allows us to take the trajectory context into consideration, which reward engineering does not allow for. For example, in the code generation task, we would like the reward model to give a higher score to a program if it is almost correct compared to when it is completely incorrect. This can only be done by using the trajectory as the reward model input. Also, for code generation we fine tune a pre-trained model to obtain our reward model, allowing us to transfer domain knowledge obtained through pre-training into the reward model.
> - However, we believe that the main novelty and contribution of our work comes from how we use the reward factors to elicit preference labels. Once we have these labels, any technique can be used to learn the reward function, including linear regression with the weak reward factors as inputs.
> - To address your second point, we remark that our algorithm is not meant for the setting where a human annotator is used to compare each trajectory. Therefore we do not include RLHF in our main experiments. However, we do include a comparison of HERON to RLHF in Appendix F.
>
> **Question 1: Problem setting description**
> - We include a description of our problem in Section 3.1 of the submitted manuscript.
> - We do not make any strong assumption on the ground truth reward, but we do expect that the hierarchical comparison rule created by HERON will be aligned with the comparison rule induced by the ground truth reward. We believe that this assumption is reasonable, as the human overseer has ranked the reward factors by importance.

---

> > ### Author Response · Authors · 2023-11-20
> > **Rebuttal Continued**
> >
> > **Minor Point: Why GT performs worse than other methods**
> > - We believe that GT performs worse than other methods due to reward shaping [1]; although the GT reward is optimal, it may not always provide enough information during training, therefore resulting in worse performance.
> >
> > Ng, Andrew Y., Daishi Harada, and Stuart Russell. "Policy invariance under reward transformations: Theory and application to reward shaping." Icml. Vol. 99. 1999.
> >
> > **Minor Point 2: Flexibility compared to Reward Engineering**
> > - Thank you for the suggestion. We believe that reward engineering and HERON provide two different types of flexibility: reward engineering will be able to better model rewards that are linear in the reward factors, while HERON will be better suited to model highly non-linear rewards.
> > - In our experiments, we find that HERON, although only using rankings over reward factors, is still quite flexible. This indicates that the fine-grained control RLHF allows may be redundant. HERON exploits this phenomena, allowing for reinforcement learning with less feedback. We can increase the flexibility of HERON by changing the margin parameter on each reward factor, therefore decreasing or increasing its significance in HERON’s preference elicitation process. We have added such discussion to the conclusion of the paper.
> >
> > **Question 2: Distinction from reward engineering.**
> > - We describe classical reward engineering and its drawbacks in paragraphs 2 and 3 of our introduction.
> > - Concretely, reward engineering assumes the ground-truth reward can be formed as a simple combination of rewards factors, an assumption that may suffer from large approximation bias. In addition, reward engineering usually requires a large tuning cost, as the optimal reward weights are not necessarily intuitive to human overseers and therefore must be chosen through trial and error.
> > - In contrast, HERON does not require a human to assign a weight to each reward factor, and instead can learn a reward from a simple ranking of reward factors. In addition, the decision tree used by HERON can be highly non-linear, possibly allowing for more complex rewards to be learned. Finally, HERON allows for the context (trajectory information) to be taken into account during the reward modeling process, which reward engineering does not.
> >
> > **Question 3: Comparison to RLHF**
> > - We emphasize that the goal of this work is not to replace RLHF, and that HERON and RLHF consider two different settings. We include a comparison to RLHF based learning in Appendix F, where we further examine the difference between HERON and RLHF.
> > - In this setting we find that HERON can outperform RLHF, as the reward factors are quite informative. We remark that HERON and RLHF are two different algorithms, although they have a similar structure. HERON requires more information than RLHF (in the form of reward factors) but less information than reward engineering.

---

### Official Review · Reviewer_yRzv · 2023-11-01

**Soundness:** 3 good
**Presentation:** 3 good
**Contribution:** 2 fair
**Rating:** 3
**Confidence:** 4

**Summary:**

This paper addressed the reward design problem in RL. This paper proposed HERON, a novel RL framework that utilizes rankings of reward factors to design reward functions by employing a hierarchical decision tree for trajectory comparisons and preference-based reward modeling. This paper assumed weak preference supervision where there are three labels in comparison: A is better, B is better, and tie. If the comparison result is "tie", then, this method moves to the next stage and does the same thing. This decision tree is a cascade of binary classifiers. In experiments, HERON outperforms reward engineering approaches, reduces tuning costs, improves accessibility, and achieves robustness in various RL environments.

**Strengths:**

- HERON introduces an approach to distill rewards from relative feature rankings, filling a crucial gap in RLHF.
- The paper provides some evaluation, spanning a wide range of tasks.
- Addressing the challenge of reducing the human labeling burden in RLHF, HERON offers a solution that achieves parity with RL in various tasks, making it a valuable contribution to the field.

**Weaknesses:**

- Marginal Improvement: HERON demonstrates only marginal or no substantial improvement over straightforward heuristic baselines in many environments, reducing confidence in its efficacy.
- Unconvincing Comparison: Directly comparing HERON to standard RLHF methods, which often operate in settings with unavailable or human-annotated reward factors, may not provide a fair or meaningful benchmark.

**Questions:**

1. Performance Comparison: How does the proposed method compare to existing state-of-the-art approaches in terms of performance, and what are the key differentiators? Comparative analysis is crucial to demonstrate the novel contributions of the proposed method. Evaluating its performance against existing methods (such as PEBBLE or Meta-RewardNet in control domain) highlights its strengths and areas where it excels. I think that comparison with existing RLFH methods is important since this paper targets the same problem in RL. (If this paper targeted only NLP domain or focused on the contribution in code generation, I think it is not essential.)

2. Addressing Limitations: Have the authors considered potential limitations and challenges of their method? Are there opportunities for further exploration or improvements in addressing these limitations?

3. Practical Significance: Is there a clear motivation and real-world application for the proposed research? How does the paper demonstrate the practical significance of the findings?

---

> ### Author Response · Authors · 2023-11-20
> **Thank you for the detailed review!**
>
> Dear Reviewer yRzv:
>
> Thank you for the detailed review of our work. We believe that your two main concerns on marginal improvements are addressable. As for the concerns about unconvincing comparisons, we believe there is a major misunderstanding. Our method is inspired by RLHF, but is not an RLHF method. We attempt to clarify in our response and in the updated paper. Please let us know if you have any further questions.
>
> **Weakness 1: Marginal Improvements**
> - We would like to point out that the improvements over comparable baselines are not marginal. In robotics, HERON results in a 7.8% improvement compared to reward engineering. For traffic light control, HERON beats all other baselines by 33%. For Coding, HERON can beat the CodeRL method by 3.9% as measured by Pass@K and by 5.2% as measured by filtered Pass@K, while greatly reducing the tuning cost. These improvements are displayed in Tables 1, 2, and 3 and Figure 3c of the submitted manuscript.
> - Moreover, we would like to point out that our method is designed to learn from rankings, greatly reducing the tuning cost of reward design. Therefore the goal of our experiments is not necessarily to beat carefully tuned rewards, but to show that we can match them while only using rankings over reward factors.
>
> **Weakness 2: Unconvincing Comparison**
> - We believe there is a misunderstanding here. In our main experiments, we do not compare our method to RLHF techniques as RLHF considers a completely different setting than our paper considers (see section 3.1 for our problem description). Instead, RLHF serves as the motivation for our HERON algorithm, as mentioned in the introduction. HERON is designed as an alternative to reward engineering.
> - We primarily compare our method with (1) policies trained with the ground truth reward (2) heuristic algorithms that receive the same amount of information of HERON (see the description of baselines on page 6) and (3) State-of-the-art reward engineering approaches for the Coding task. All of these algorithms either receive more or an equal amount of information in the reward design process than HERON, and are therefore meaningful baselines.
>
> **Question 1: Comparison with RLHF Baselines**
> - As we previously mentioned, our work is inspired by RLHF, but is not designed for the RLHF setting. Therefore it would not be meaningful to compare it to methods like PEBBLE and MetaReward-Net, which we consider to be orthogonal directions of work. However, we do include a comparison to RLHF based learning in Appendix F, where we further examine the difference between HERON and RLHF.
>
> **Question 2: Addressing Limitations**
> - HERON essentially proposes a new RL setting, where only weak feedback is given. The main limitation of HERON is that not every RL environment will contain an obvious ranking over the reward factors, as some factors may be equally important. We have added more discussion in Appendix L of the updated paper.
>
> **Question 3: Practical Significance**
> - Our main motivating problem is code generation, where reward engineering may suffer from approximation bias and require a large tuning effort. In contrast, ranking the reward factors will be easy. Concretely, CodeRL introduces 4 tunable reward parameters, while PPOCoder has 6. In this task, we show that HERON can achieve significantly better performance than state-of-the-art baselines, therefore demonstrating HERON’s practicality in the very useful task of code generation. Similarly, we achieve great results in traffic light control, which is a practical and real world application.

---

### Official Review · Reviewer_rUJV · 2023-11-01

**Soundness:** 3 good
**Presentation:** 2 fair
**Contribution:** 2 fair
**Rating:** 5
**Confidence:** 3

**Summary:**

In reinforcement learning, it is crucial to design a reward function that reflects the objective of the task. It can be complicated or error-prone to define the exact reward function. In practice, we can define some reward factors and the reward function is a weighted combination of the reward factors.

Instead of using a weighted sum of reward factors, this work proposes a ranking approach, where the reward function is determined by the ranking of the reward factors. The HERON framework is evaluated on various domains.

**Strengths:**

Compared to weights, ranking can be more interpretable and easier to design. Empirically, RL trained using the ranking-based reward model outperforms other baselines.

**Weaknesses:**

Rankings of reward factors can be less expressive than weighted combinations of reward factors. In other words, some reward functions can be expressed as weighted sums, but not as a ranking of reward factors. It would be helpful to have discussions on this.

Clarity: It would be helpful to make the setting clearer. My understanding is that the input to the reward model training process is no longer preference data ($ \{ (\tau_w, \tau_l), \dots \} $), but an ordered list of reward factors. In other words, the reward model training part is not changed, but only the input to the reward model training is changed.

Minor:
* In Alg 1, the condition in while is not syntactically correct ($l \leq n \ \mu = 0$).
* You may move the legend in Fig. 3 out of the plot so readers can see the plot clearly.

**Questions:**

In the experiments, if the ground-truth reward function is a linear combination of reward factors, using the rank of significance of the reward factors is enough to learn a good policy? Empirically, this seems to be the case for robotic control tasks. Can authors provide an intuition behind this?

Can reward factors serve as a reward model directly without using the method proposed in this paper? For example, we can define a reward function like 10^4 * first_reward_factor + 10^3 * second_reward_factor + ..., where 10^i can be adjusted.

---

> ### Author Response · Authors · 2023-11-20
> **Thank you for the helpful review!**
>
> Dear Reviewer rUJV,
>
> Thank you for the thoughtful review and suggestions. We provide our response to your concerns and questions below.
>
> **Weakness 1: Lack of expressivity.**
> - Thank you for pointing this out. We remark that HERON and classical reward engineering are two different design principles, and which one can better capture the reward may be problem specific. However, decision trees can be highly non-linear, therefore allowing HERON to approximate a wide variety of reward functions.
> - In practice, we typically find that the HERON framework does provide enough expressivity, as shown by Section 4.3 where switching the order of secondary and tertiary rewards correctly changes how the agent acts. We can further increase the flexibility of the HERON framework by tuning the margin parameters. For example, if we would like to increase the emphasis of the reward on a given reward factor, we can decrease the size of the margin for that reward factor. In this way, a greater percentage of the decisions would be taken according to the reward factor we wish to emphasize. We have added discussion about how to increase flexibility in section 4.3 of the updated paper.
> - Finally we remark that HERON enhances the capability of reward modeling in two ways: (1) it allows us to take advantage of the contextual information contained in the trajectories (which reward engineering does not do) by predicting the reward based on the trajectory and (2) it allows us to use pre-trained models as the initialization of the reward, therefore vast amounts of domain knowledge (e.g., semantic and syntactic knowledge of code) can be transferred to the learnt reward.
>
> **Weakness 2: Lack of clarity on the setting.**
> - We believe there is a misunderstanding here. The general setting we consider is one where at each time step, we receive a state observation, take an action, and then receive the next state as well as a list of reward factors. This setting is the same as that commonly used for reward engineering. Our setting is described in section 3.1 of the submitted manuscript.
> - With regards to the reward modeling setting, the only difference with RLHF comes from how we obtain the preference labels. The input to the reward model is still the trajectory segments, but now the preference label is obtained from HERON’s decision tree over the reward factors. This forms the key novelty of our algorithm. Otherwise all aspects remain unchanged from standard reward learning in RLHF. These details can be found on page 4 of the submitted manuscript .
>
> **Minor comments.**
> Thank you for pointing out the error in Algorithm 1 and suggesting improvements for Figure 3a. We have fixed the algorithm in the updated paper and edited Figure 3a to make the curves more visible.

---

> > ### Author Response · Authors · 2023-11-20
> > **Rebuttal Continued**
> >
> > **Q1: Robots intuition**
> > - Our intuition is that although the reward learned by HERON may be slightly misaligned with the ground truth reward, it can provide a good approximation to the ground truth reward. In fact, there may be multiple reward functions that lead to good performance: HERON provides a principled way to find one while only using ranking over reward factors.
> > - We find that the performance of HERON varies on a case by case basis: In environments where the linearity of the reward function is important it may not do as well. However, in all of our experiments we find that HERON is able to perform at least similarly to reward engineering if not better.
> >
> > **Q2: New baseline**
> > - We have included this exact baseline in the submitted manuscript. Please see the paragraph on baselines in Section 4.1. We formulate the reward as $\sum \beta^i z_i$, where $z_i$ are the reward factors and $\beta$ can be tuned. More detailed analysis of this baseline can be found in appendix C.
> > - We remark that two advantages of HERON over the mentioned baseline is that it can take into account contextual information from the trajectory, and it can be initialized with a pre-trained model to take advantage of pre-trained knowledge.

---

### Official Review · Reviewer_9kSM · 2023-11-01

**Soundness:** 3 good
**Presentation:** 3 good
**Contribution:** 2 fair
**Rating:** 5
**Confidence:** 5

**Summary:**

The paper introduces HERON, which leverages the rankings of multiple reward factors (objectives) to derive reward functions using a hierarchical decision tree. This paper is focused on scenarios where numerous reward factors are available for every state-action pair, and it assumes that human experts have ranked these factors to establish their relative importance (weak preferences). When comparing two trajectories, HERON systematically evaluates the reward factors in sequence to determine their preferences. To elaborate, HERON initiates the comparison with the first reward factor, and the labeling procedure (i.e., assigning a binary label) concludes if the disparity exceeds a predefined threshold. If the dissimilarity falls below the threshold, HERON proceeds to compare the next reward factor, iteratively following this process until all the reward factors have been assessed.

**Strengths:**

* Paper is well structured and easy to follow.

* Extensive evaluation: The authors have extensively validated their method across a wide selection of tasks, including classic control, robotic control, multi-agent traffic light control, and large language model fine-tuning for code generation.

**Weaknesses:**

* Lack of justification in reward learning from preferences. In this work, based on ranking between multiple objectives, the authors first generate preferences and train a reward function using cross-entropy loss, which stems from the Bradley–Terry model. Here, the motivation for utilizing a preference-based reward learning framework is unclear. In the original preference-based RL framework, human preferences are generated from the Bradley–Terry model under an unknown utility function and the goal of reward learning is approximating this function based on preference datasets. However, when we have a ranking between multiple objectives, why we also use this framework is unclear. Basically, what is the target of the reward learning? The authors need to clarify this part.

* Lack of ability in personalization. Unlike the standard preference-based learning framework, the ability to obtain personalized rewards is limited. Humans only can specify the ranking between objectives and it is hard to control the tradeoff between objectives. For example, let's consider a case where there are two objectives. Assume that there are two human annotators with the same ranking but the first human wanna get a reward function that emphasizes the first objective more. In this case, this framework can't provide a different reward for two human annotators (because the ranking is still the same). It would be nice if the authors could discuss this limitation.

**Questions:**

* Learning curves on robotics environments. It would be nice to include the learning curves on robotics environments in the main draft or appendix.

* Standard deviation across different trials is quite high in Figure 3a and Figure 3c. It would be nice to tune some hyper-parameters for DDPG or Q-learning.

---

> ### Author Response · Authors · 2023-11-20
> **Thank you for the detailed review!**
>
> Dear Reviewer 9kSM,
>
> Thank you for the insightful and clear review! We provide our responses to your questions and concerns below and look forward to further discussion.
>
>
> **Weakness 1: Lack of clarity regarding motivation of reward modeling.**
> - Note that we have explained our motivation of employing reward modeling in paragraph 5 of the introduction, where we state that we use a reward model to leverage human rankings over reward factors. Here we would like to summarize our motivation again:
>
> - The main motivation of HERON is to replace reward engineering, which requires handcrafting the reward. HERON uses human rankings over reward factors to automatically generate preferences between trajectories. These comparisons are then used to train a reward model. The motivation of using a reward model is to convert the preferences between trajectories into a numerical reward that can then be used to train the policy.
> - In the updated version we have added a clarifying sentence to paragraph 5, stating “The reward model provides us a principled manner to convert rankings over the reward factors into a numerical reward.”
>
> **Weakness 2: Lack of ability in personalization.**
> - This is an interesting point. Beyond changing the ranking, we can increase the flexibility of HERON by tuning the margin parameters in the HERON framework. For example, if we would like to increase the emphasis on a given reward factor, we can decrease the size of the margin for that reward factor. In this way, a greater percentage of the decisions would be taken according to the reward factor we wish to emphasize. In the updated version, we have added discussion about how to increase flexibility in section 4.3, writing “To further increase the flexibility of HERON, we could tune the margin parameter of each reward factor, therefore impacting the fraction of decisions made at each level of the decision tree.”
> - In addition, in the case where we have two factors, we can randomly drop the comparison based on the second factor is we would like to increase the dependence on the first factor.
> - Finally we remark that HERON has two additional benefits for enhancing the capability of reward modeling, which the linear combination of the reward factors does not have:
> (1) HERON allows us to take advantage of the contextual/state information contained in the trajectories by predicting the reward based on the trajectory and (2) for certain applications, e.g. code generation, it allows us to use pre-trained models as the initialization of the reward, and therefore vast amounts of domain knowledge (e.g., semantic and syntactic knowledge of code) can be transferred to the learnt reward.
>
> **Question 1: Learning curves in robotics environments**
> - We have included the curves in Appendix K of the updated paper. From these curves, we observe that HERON matches or beats the performance of reward engineering while being much more stable.
>
> **Question 2: Hyperparameters in Figures 3a and 3c.**
> - We have sufficiently tuned the baselines in both of these experiments, selecting the best policy (over 5 random seeds) from the learning rates {1e-6, 1e-5, 1e-4, 1e-3}. We believe that these two baselines have high variance due to the poor shaping of the reward, which can lead to instability in training.

---

> > ### Comment · Reviewer_9kSM · 2023-11-21
> >
> > Thank you for your answers but my concerns on problem setup are not fully addressed.
> >
> > For personalization, tuning margin parameters makes sense but why not adjust weights (over multiple objectives) directly instead?
> >
> > This is also relevant to my question about clarity: why do we need to learn reward function from preferences based on ranking between objectives?
> >
> > This setup is closely related to multi-object RL setup, and there are many works in that direction.
> >
> > However, I feel like the authors didn't discuss the connection (and differences) with multi-object RL and this makes it hard to understand the necessary (or practicality) of the problem setup.
> >
> > After reading the rebuttal and other reviewers' comments, my impression is that other reviewers also have a similar concern on the problem setup and the motivation of this work is not clear.
> >
> > Due to this, I'd like to keep my score as borderline reject.

---

> > > ### Author Response · Authors · 2023-11-21
> > > **Thank you for the fast reply!**
> > >
> > > Dear Reviewer 9kSM,
> > >
> > > Thank you for the fast response to our rebuttal. We respond to your questions below.
> > >
> > > **For personalization, tuning margin parameters makes sense but why not adjust weights (over multiple objectives) directly instead?**
> > > We remark that HERON and reward engineering are two different design approaches, and which one works better will depend on the problem. However, when domain knowledge such as the importance ranking of the reward factors are available, HERON provides a much more intuitive way to do reward modeling than heuristic reward engineering. Note that the importance ranking of reward factors does not necessarily help determine their reward engineering weights, as the reward factors could be correlated, the scale of the reward factors may be different, and the reward factor scales could even change throughout training. HERON allows us to bypass or mitigate all three of these concerns.
> > >
> > > **This is also relevant to my question about clarity: why do we need to learn reward functions from preferences based on ranking between objectives?**
> > > The reward function is a critical part of HERON. We propose to learn a reward function as it allows us to design a reward from an intuitive form of domain knowledge (importance rankings). Learning a reward function also gives us robustness to training domain shifts, allows us to take into account the contextual/state information contained in a trajectory, and it allows us to transfer pre-trained knowledge (like CodeT5 for code generation) into the reward function.
> > >
> > > **This setup is closely related to multi-object RL setup, and there are many works in that direction.**
> > > We want to clarify that this paper is not focused on the multi-objective RL problem. Multi-objective RL typically tries to find a set of Pareto optimal policies [1]. In contrast, all of our experiments focus on the setting where there is one most important objective (such as passing all unit tests for code generation or throughput in traffic light control). An interesting line of future work would be to explore the potential applicability of HERON to learn optimal policies for each possible reward factor ranking, but it is beyond our current scope in this paper.
> > >
> > > [1] Van Moffaert, Kristof, and Ann Nowé. "Multi-objective reinforcement learning using sets of pareto dominating policies." The Journal of Machine Learning Research 15.1 (2014): 3483-3512.
> > >
> > > **However, I feel like the authors didn't discuss the connection (and differences) with multi-object RL and this makes it hard to understand the necessary (or practicality) of the problem setup.**
> > > In the latest revised manuscript, we have already added the discussion about the connection and differences with existing work on multiobjective RL. However, we would like to emphasize again that we consider the same settings as those in existing work, where reward engineering has been widely adopted.
> > >
> > > Thank you again for your thoughtful questions. Please let us know if you have any other concerns.

---

### Meta-Review · Area_Chair_18JS · 2023-12-06

**Metareview:**

This paper develops a hierarchical deep reinforcement learning method with weak preference feedback called HERON.

HERON uses a hierarchical decision tree to compare trajectories based on the most important factors, and then proceeds to evaluate less important factors. It does not rely on the magnitude of each reward signal and allows the design of reward functions without specifying numerical weights for each reward factor.

A typical accepted reinforcement learning paper has either rigorous theoretical analysis or concrete experimental results. For deep RL papers, one would expect novel ideas/clear experimental advantages.

There is a line of work on preference-based reinforcement learning that is closely related to the proposed method. The AC would suggest that the authors read the research and review papers on Preference-Based Reinforcement Learning.

**Justification For Why Not Higher Score:**

The AC agrees with all reviewers that this paper needs significant improvement before it can be accepted.

**Justification For Why Not Lower Score:**

N/A

---

### Decision · Program_Chairs · 2024-01-16

Reject